

# Novel decorating behaviour of silk retreats in a challenging habitat

Alfonso Aceves-Aparicio[1,2,3], Donald James McLean[1], Zoe Wild[1],
Jutta M. Schneider[2] and Marie E. Herberstein[1]

[1] School of Natural Sciences, Macquarie University, Sydney, New South Wales, Australia
[2] Institute of Zoology, Universität Hamburg, Hamburg, Germany
[3] Max-Planck-Institut für Chemische Ökologie, Jena, Thüringen, Germany

## ABSTRACT

Many ecological interactions of spiders with their potential prey and predators are affected by the visibility of their bodies and silk, especially in habitats with lower structural complexity that expose spiders. For instance, the surface of tree trunks harbours relatively limited structures to hide in and may expose residents to visual detection by prey and predators. Here we provide the first detailed description of the novel retreat building strategy of the tree trunk jumping spider *Arasia mullion*. Using fields surveys, we monitored and measured over 115 spiders and 554 silk retreats. These spiders build silk retreats on the exposed surface of tree trunks, where they remain as sedentary permanent residents. Furthermore, the spiders decorate the silk retreats with bark debris that they collect from the immediate surrounding.
We discuss the role of silk decoration in the unusual sedentary behaviour of these spiders and the potential mechanisms that allow *A. mullion* to engineer their niche in a challenging habitat.

## INTRODUCTION

Habitat structural complexity (the quantity, composition, and spatial arrangement of biotic and abiotic elements) can be a major challenge for some animals. Consequently, the degree of complexity influences strategies that maximise fitness (*Gigliotti et al., 2020*). Habitats with relatively high complexity can concentrate resources and offer greater cover that reduces encounter rates with predators at higher trophic levels (*Langellotto & Denno, 2004*). In comparison, less complex habitats can be challenging for animals as both food and cover from predators are less abundant. Even in low complexity environments, some animals can overcome these habitat limitations through building behaviour, also known as 'extended phenotype'. This refers to traits expressed beyond the boundaries of the animal's body, such as building constructions to hide under, hunt with or breed in (*Dawkins, 1982*).

Spiders are well known for their conspicuous expression of extended phenotypes through their web building behaviour. These silk constructions overcome vegetation gaps up to 10 m (*e.g.* across water bodies: *Gregorič et al., 2011*). Similarly, the large and complex

Corresponding author
Alfonso Aceves-Aparicio,
bioarach@gmail.com

three-dimensional colony webs cover large portions of trees and create a foraging, mating and communication platform for the entire colony of spiders (*Nentwig, 1985*; *Eberhard, Agnarsson & Levi, 2008*). In highly exposed habitats, spiders deploy silk to reduce the risk of predation. For example, some aerial webs include silk scaffolding to detect or deter approaching predators, such as wasps or birds (*Cloudsley-Thompson, 1995*). Leaf curling spiders wrap a leaf into the centre of the web into which they retreat instead of sitting at the exposed web hub (*Thirunavukarasu, Nicolson & Elgar, 1996*). Silk constructions can also be used to facilitate the exploitation of otherwise challenging environments. For instance, buckspoor spiders of the genus *Seothyra* shelter from the extreme thermal conditions in the Namib desert dunes (up to 73 °C on the sand surface) in underground burrows lined and covered with silk (*Lubin & Henschel, 1990*). Spiders of the genus *Wendilgarda* attach sticky silk lines to the surface of water streams to capture insects that move on the water surface (*Coddington & Valerio, 1980*). Finally, the unique diving bell spider (*Argyroneta aquatica*) takes the occupation of inhospitable environments even further by building a silk nest underwater, where it lives its entire life (*Seymour & Hetz, 2011*).

Considering the functionality, flexibility, and broad benefits of web building, it is surprising that some groups have abandoned capture webs and instead hunt by vision and ambush (*e.g.*, jumping spiders) (*Richman & Jackson, 1992*; *Hill & Richman, 2011*; see also *Wolff, Wierucka & Uhl G.Herberstein, 2021*). Yet, they may still utilise silk for overcoming some habitat constraints, in the form of building retreats or abseiling with a dragline (*Herberstein, 2011*). Jumping spiders (Salticidae) are the most speciose family of spiders with 6,344 known species (*World Spider Catalog, 2021*), of which most are cursorial predators inhabiting virtually all terrestrial ecosystems (except for the poles). Given the astoundingly acute vision of jumping spiders, they visually assess the complexity of their surrounding environment (*Jackson & Blest, 1982*; *Aguilar-Argüello, Gerhard & Nelson, 2019*) and navigate complex habitats when searching for prey or mates. Some jumping spiders build and occupy hidden silk retreats (*e.g.*, underneath leaves or behind bark or rocks) when inactive or when reproducing (*Richman & Jackson, 1992*; *Hoefler & Jakob, 2006*).

When exploring less complex and more exposed environments like tree trunks, most species of jumping spiders rely on rough, highly contrasting surfaces for concealment (*Cumming & Wesołowska, 2004*; *Robledo-Ospina et al., 2017*). Thus, we would not expect jumping spiders to be generally active on, nor to inhabit, highly exposed habitats such as smooth and bright bark. Based on preliminary surveys of spiders occupying tree trunks, we became interested in the tree trunk jumping spider *Arasia mullion* (*Zabka, 2002*) for several reasons. First, these spiders occurred in unexpected high abundances on tree trunks. Second, unlike other spiders that inhabit tree trunks, *A. mullion* was mostly found occupying a silk retreat during the day, which is uncommon for jumping spiders. This species exhibits a prolonged reliance on using a multi-purpose silk retreat on the exposed surface of trunks of a small range of tree species. The trees occupied by these spiders seem unlikely permanent niches for animals as their smooth and bright surfaces are likely to render residents highly conspicuous.

These observations generated a number of research question that we wished to address: (1) how do these spiders build their silk retreats; (2) what is the overall phenology of this species and how consistently are silk retreats occupied; (3) how do they decorate their retreats and how does decoration affect the visual quality of the silk? Here, we provide the first detailed descriptions of *A. mullion's* retreat building behaviour and discuss the natural history of the retreats in this challenging environment.

## METHODS

### Study species

*Arasia mullion Zabka, 2002* are recently discovered small jumping spiders (Salticidae) that are locally abundant on trees in eastern Australia. Currently, they are thought to be endemic to New South Wales (*Zabka, 2002*) and very little is known about their behaviour, life history, or the characteristics of the habitats they occupy.

### Study site

Observations were performed in approximately 6,500 m$^2$ on the Macquarie University campus (Macquarie Park, NSW, Australia). Within our study site, 57 trees were inspected for the presence of spiders. Every tree was identified to species and individually labelled. A total of 19 trees were periodically monitored for 16 months to observe the spiders' natural history, including activity patterns, retreat construction and occupancy, and inter-and intra-specific interactions (detailed below).

### Retreat building

*Arasia mullion* spiders are commonly found in their silk retreats; close observation of the silk retreats revealed the presence of fine debris attached to the silk. To understand the retreat-building behaviour and determine the origin of the debris on the silk, 20 spiders were captured, and their retreats were removed from the tree trunk surface. After 1 h, spiders were released back onto their original trees, then periodic observations were conducted four times a day for 5 days to observe the building of new retreats. Four instances of retreat building behaviour were filmed using a compact camera (Samsung TG-5, 12MP) on a tripod or with a hand-held mobile phone camera (Apple iPhone 6S).

### Phenology and retreat occupancy

We conducted phenological surveys to determine habitat occupancy patterns (if any) throughout the seasons. Six surveys were conducted at approximately three-month intervals between July 2018 and November 2019. During each survey, all retreats detected on the surface of the tree trunks from the ground up to a height of 2 m were labelled with permanent marker; labels were placed 10 cm from the retreat to reduce the possible effect of increased visibility or conspicuousness. As the retreats appeared to be sessile structures, we considered all the retreats we found without a label code during subsequent surveys to be newly built. All newly built retreats were then labelled according to the chronological survey number. All retreats were photographed with their labels and a scale for reference (Fig. 1).

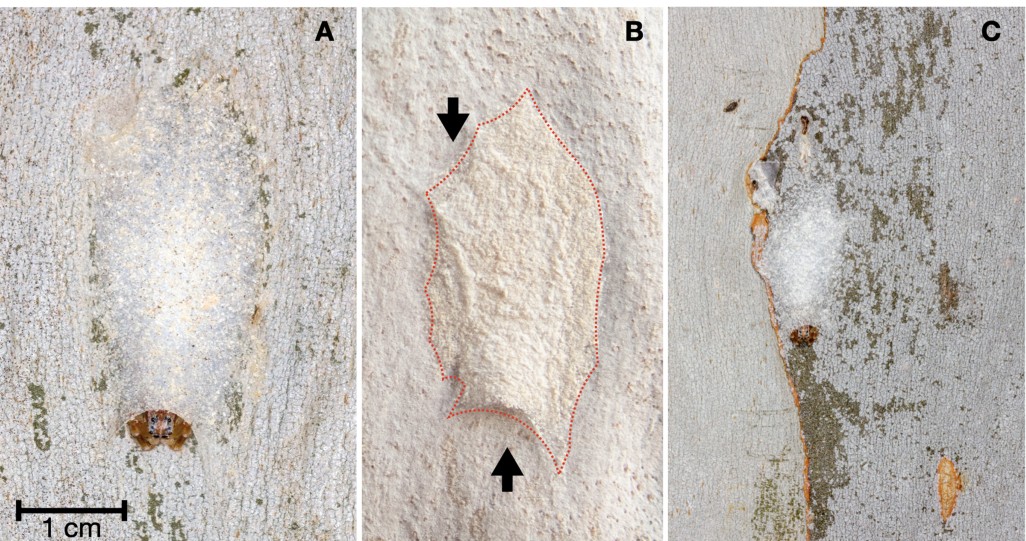

**Figure 1** **Typical appearance of the silk retreat of *Arasia mullion* and its position on a tree trunk.**
(A) The prosoma and legs of *A. mullion* are shown protruding from the bottom opening of the
retreat. (B) Silk retreat of *A. mullion* which visually matches the appearance of the tree trunk surface of
Sydney Blue Gum (*Corymbia maculata*). The dotted line outlines the edge of the silk retreat. The black
arrows point to the top and bottom openings. (C) Frontal view of a silk retreat on the trunk's surface
where scarring from debris collection is visible. Scars occurred in the immediate surroundings of the
retreat and commonly stretched outwards from both openings. Photo credit: Alfonso Aceves-Aparicio.

All marked retreats were checked for spider presence by gently depressing the silk from
the top edge down with a paintbrush which caused the resident spider to emerge at the
lower opening. A subset of spiders were collected in vials or photographed *in situ* with a
scale, and we measured the prosoma width and length from the photographs. Photographs
of retreats were manually converted to monochrome retreat outlines using ImageJ
version 1.52k (*Schneider, Rasband & Eliceiri, 2012*). Retreat dimensions were measured
from these outlines. Outline images were smoothed by applying Gaussian blur
(sigma = 10). Retreat width was calculated as the length of the longest row of pixels in the
outline, and length was the longest column. Lastly, area was the number of pixels
within the outline shape. Width, length and area were scaled to convert pixels into mm or
mm$^2$. All statistical analyses, including log transformations to meet statistical assumptions,
were conducted in R (*R Core Team, 2021*). Unlike silk retreats, individual spiders
could not be marked without compromising its appearance and thus survival against
predators. Therefore, for this study we considered that it was highly unlikely to measure
the same individual although this cannot be excluded entirely.

## Silk reflectance—comparison with unusual samples

A small group of *A. mullion* spiders was found living on a concrete water tank in a small
area of public parkland (3,000 m$^2$, North Ryde, NSW, Australia, 33°48′02″S, 151°08′13″E).
They were part of a small local population that mostly occurred on trees. The retreats
built on concrete lacked the debris normally covering tree retreats, which provided us with

an unusual opportunity to compare retreats with and without debris. All the retreats found on the concrete tank were collected, $n$ = 5 (non-decorated samples), and six silk retreats were collected from nearby tree trunks for comparison (decorated samples). Samples were then placed flat over black cardboard to measure their reflectance spectra. We used spectrometry to assess potential visual differences between these samples resulting from the lack of debris decoration. The reflectance spectra measurements were obtained using a Jazz Ocean Optics spectrophotometer (Ocean Optics, Largo, FL, USA) with the following settings: integration time = 40 ms, boxcar width = 10, averaged scans = 10. We used a PX-2 pulse xenon light source and all measurements were relative to a white standard WS-1. The light source and probe were set at an angle of 45 degrees to the cardboard. Measurements were restricted to the UV and visible part of the spectrum of light (between 300 and 700 nm) as this range is relevant to most ecological observers of spiders (*Cronin et al., 2014*). Each measurement was taken five times, and the results averaged. All spectral processing and exploration were carried out using the R package Pavo in R, versions 2.2 and 3.5.2, respectively (*Maia et al., 2019*; *R Core Team, 2021*).

## RESULTS

### The unusual sedentary life of a jumping spider

We found that *Arasia mullion* led a sedentary life on the surface of tree trunks, where they build, decorate, and occupy their silk retreats (described in detail below). Of the 21 species of trees within our study area, *A. mullion* was restricted to the following four species: Spotted Gum (*Corymbia maculata*), Smooth-barked Apple Myrtle (*Angophora costata*), Flooded Gum (*Eucalyptus grandis*) and Scribbly Gum (*Eucalyptus racemosa*). The traits that these tree species have in common are smooth bark with small depressions ("dimples") on the surface, which the spiders used as the foundation of their retreats. The remaining 17 species have bark characterised by fissured, fibrous, stringy textures or shedding in irregular flakes, which creates rough surfaces with deep elongated crevices where we did not record any *A. mullion* retreats (Table S1).

### The silk retreats

The retreats of *A. mullion* were exclusively built over a dimple on the exposed surface of tree trunks. The retreat consisted of a sheet of silk laid on the surface of the trunk (Figs. 1A, 1B). The retreats varied in colour and visual texture. However, each retreat matched the bark on which it was situated (at least to the human eye). Each silk retreat had two openings, one at the top and one at the bottom. When active, the spider sat with only the front part of the body and two pairs of legs protruding from the lower entrance (Figs. 1A, 1B).

### Retreat building behaviour

The fact that the colour and texture of the silk retreats matched their backgrounds was immediately noticeable. However, whether this was achieved passively (debris caught in the silk) or actively (spiders decorating their retreat with debris) was unclear. Despite the construction of new retreats being relatively uncommon to observe (compared to the

number of retreats and spiders present), we were able to film the construction behaviour of a number of retreats after a spider had been removed and replaced (Video S1).

The spiders commenced construction by laying fine silk lines over a suitable dimple on the surface of the tree trunk. Then, using their chelicerae, they scraped the tree surface and collected fine debris. We observed debris being collected from the immediate surroundings of the retreat and from further away. After collection, the debris was carried back to the dimple underneath the silk lines. The spiders then used the debris to decorate the silk by brushing it onto the underside of the silk using active movements of their pedipalps. Once the debris had been applied, they added several layers of silk to the underside of the retreat using oscillatory movements of the spinnerets. These layers attached the debris to the initial silk lines. This process was conducted repeatedly (Video S1). Although we were not able to document the building of these retreats from start to finished, our observations suggest that this can vary between 2 and 4 h.

The process of scraping debris left visible scars on the bark, which allowed us to confirm that the observed decoration behaviour occurred in all of the surveyed retreats, even when we had not directly observed retreat construction. The scarring from the debris collection also allowed us to confirm that the spiders use debris directly from the dimple and the immediate surroundings of the retreat (Fig. 1C).

## Phenology

The jumping spiders *A. mullion* spent their entire life on the surface of tree trunks. Unlike other salticids *A. mullion* spiders stayed in their silk retreats and were rarely seen wandering the surface of the tree trunks. The retreats were occupied by all life stages, from recent hatchlings to mature males and females. Our surveys showed an annual life cycle with egg sacs laid inside the retreats in late Australian spring and early summer—November to December. Early instars were found in newly spun retreats during January and February. Our surveys covered two phenological cycles with two surveys conducted during 2018 and four during 2019. Size variation in both the spiders and their retreats was recorded during each survey (Fig. 2).

We assessed the effect of occupancy (vacant and occupied) and time (yearly season) on the recorded size of the available retreats (length) using a linear mixed effects model. We set occupancy, season and their interaction as fixed effects and the individual ID of each silk retreat as a random effect. The length of silk retreats was log-transformed for normality (Shapiro-Wilk test, $p = 0.44$) and to meet the homogeneity of variance assumption (Levene's F test, $p = 0.282$).

Overall occupied retreats were larger than vacant retreats. Retreat size significantly varied between the seasons, increasing in size from summer to spring. The interaction of occupancy and season indicates that in summer occupied and vacant were similar in size whereas in autumn, winter and spring occupied retreats were larger than vacant retreats (Table 1, Fig. 3). The size increment of the silk retreats was analysed separately for newly recorded retreats in 2019. The length of newly built silk retreats in 2019 was log-transformed for normality (Shapiro-Wilk test, $p = 0.40$) and to meet the homogeneity of variance assumption (Levene's F test, $p = 0.382$). As such, the Fisher's ANOVA was used

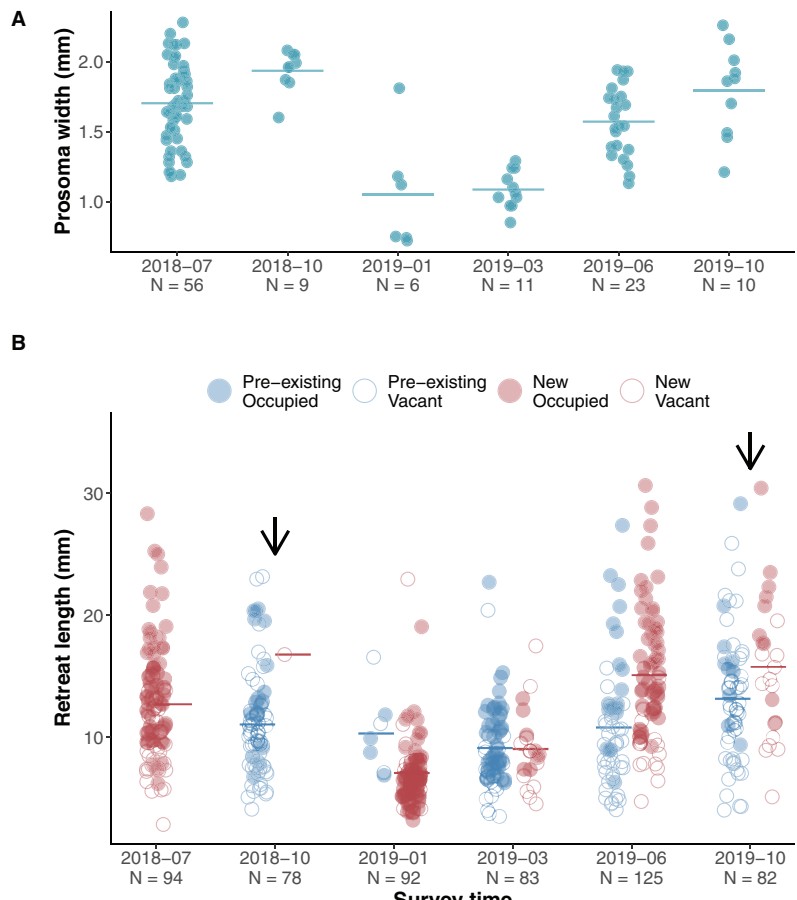

**Figure 2 Phenological size change in spiders and their silk retreats.** (A) Scatterplot of spider size during surveys, horizontal lines indicate means. The first two surveys captured size variation of subadult and adult stages at the end of the 2018 cycle, while 2019 surveys showed a clear increasing pattern as hatchlings developed. (B) Scatterplot of silk retreat size during surveys, horizontal lines indicate means. Within each survey, data points are grouped into pre-existing (blue left-hand clusters) or newly constructed (red right-hand clusters). For the first survey (July 2018), only the newly constructed cluster is shown as we account for no pre-existing retreats. Filled circles (red for newly built, blue for pre-existing) show occupied retreats while empty circles indicate that the retreat was vacant. Arrows indicate the presence of adult spiders during the survey.

to determine statistically significant differences in retreat size between surveys, F (3, 55.9) = 65.8, *p* < 0.001. The mean size of newly built silk retreats increased continuously during 2019. We used the Tukey Post-hoc test to determine which survey transitions significantly differed in retreat size (Table S3). The size increments were significant between all transitions except for the transition between June and October (2019-06 and 2019-10).

## Dynamics of retreat occupancy
### *Retreat persistence*

We recorded a total of 306 silk retreats among the six surveys (two surveys in 2018 and four in 2019). During 2018, we recorded 94 retreats in 2018-07 and only one new retreat in the following survey 2018-10. The persistence of the retreats in between transitions varied

**Table 1 Results of a linear mixed-effects model for the effect of occupancy (vacant or occupied) and season on the size (log length) of the silk retreats of *A. mullion*.**

**Mix model-fixed effect omnibus tests**

|  | F | Num df | Den df | *p* |
|---|---|---|---|---|
| Occupancy | 57.01 | 1 | 352 | <0.001 |
| Season | 40.98 | 3 | 319 | <0.001 |
| Occupancy * Season | 5.78 | 3 | 335 | <0.001 |

**Note:**
  Satterthwaite method for degrees of freedom.

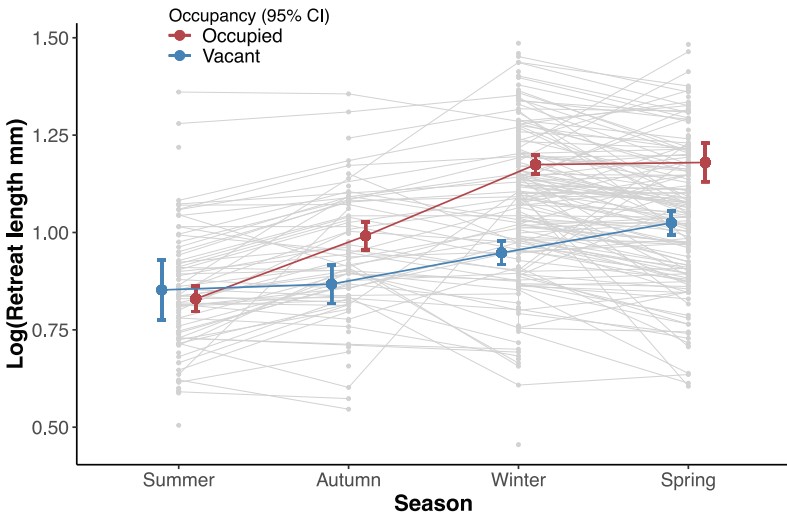

**Figure 3 Effect of the interaction between survey and occupancy status on the retreat size.** Circles denote the estimated mean retreat size (log length) for each factor and error bars are 95% confidence intervals (red colour for occupied and blue for vacant retreats). The light grey circles show the observed values for all retreats at a given season and lines are shown between circles that were recorded in more than one season.

between 47–82% for all surveys except for the transition between 2018-10 and 2019-01 where persistence was reduced to 8.75%. The disappearance of the silk retreats during this transition coincided with bark shedding by the trees in our study area. During the first survey of the 2019 new phenological cycle (2019-01), we recorded 85 new retreats of which 74.11% persisted to the next survey (2019-03), and 18.18% persisted for the remaining span of the phenological cycle to the end of 2019.

## Occupancy transitions

The persistence of an unoccupied silk retreat on the tree trunk allows other *A. mullion* spiders to find and occupy the vacant retreats. Accordingly, we recorded all the possible occupancy status transitions in retreats between surveys (*i.e.*, occupied-vacant, vacant-occupied, occupied-occupied and vacant-vacant). When spiders were pushed away from their retreats (*e.g.*, when inspecting retreats for occupants), some individuals were seen inspecting or entering a different retreat. Further, we found that persistent retreats between surveys were equally likely to increase or decrease in size independently of their occupancy

status transition (*e.g.*, remain vacant or occupied and become vacant or occupied, Table S4).

## Spectral reflectance

We used spectrometry methods to explore spectral differences between decorated and the highly uncommon undecorated silk retreats of *A. mullion* spiders (Figs. 4A, 4B). The reflectance spectrum of the silk (decorated or undecorated) showed no defined peaks between 300 and 700 nm. The undecorated silk spectra gradually decreased from 47.24% mean reflectance at 300 nm to 36.83% at 700 nm. The decorated silk gradually increased from 20.48% at 300 nm to 40.46% mean reflectance at 700 nm. The greatest difference between decorated and undecorated spectra occurs between 300 and 470 nm where there is no overlap between the reflectance values of the retreat types (decorated and non-decorated) nor the estimated SD (Fig. 4C). No meaningful statistics are included here given the small sample size of the unusual non-decorated silk retreats.

## Additional natural history notes

**Foraging.** *A. mullion* ambushed prey from inside their silk retreats, launching attacks upon visual detection of prey moving nearby. Spiders either fed on their captured prey near their retreats or returned with captured prey to the ambush position inside the retreat when prey size allowed.

**Retreat.** Upon disturbance either by approaching humans, birds, ants or large bugs, spiders moved inside their silk retreats. *A. mullion* used their forelimbs to flatten and close the entrances of their retreats.

**Agonistic interactions.** Spiders were observed performing ritualised agonistic displays against conspecifics upon visual detection. Displays included the lateral extension of frontal legs and "twisting" of the opisthosoma. Instances of this behaviour were observed when a spider attempted to enter or approach an occupied retreat. The resident spider usually confronted the intruder by maintaining an aggressive posture near the entrance of the retreat.

**Evictions.** When a spider wandering the tree trunk encountered a silk retreat, it tapped repeatedly near the openings before attempting to enter. However, when the retreat was occupied, either the resident exited immediately, or the intruding spider entered the retreat before both exited the retreat. Once outside the retreat, both spiders faced each other off as described in "Agonistic interactions". When the intruding spider was substantially larger than the resident, the resident moved away and the intruder took up position in the silk retreat (*n* = 14).

**Males' search for females.** As males reached sexual maturity, they were increasingly found wandering the surface of the tree trunks. Males were seen inspecting retreats to ultimately enter and stay in the retreats occupied by females (see below). Both instances where males were accepted or rejected were observed.

**Co-occupancy.** During spring (the mating season), several retreats were occupied by both a female and a male together. In some instances, each spider occupied one of the entrances in their common hunting position.
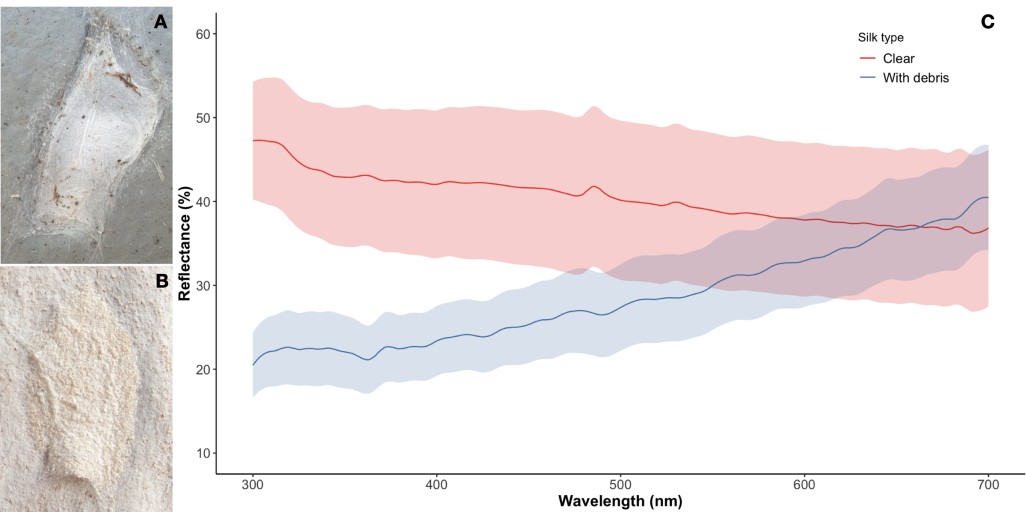

**Figure 4 Difference in appearance and reflectance spectra between decorated and non-decorated silk retreats.** (A) Silk retreat constructed on the side of a concrete water tank. The fine debris is missing while a few relatively large particles are present. (B) The usual decoration of the retreats covers the exposed silk with debris from the surrounding area of the tree trunk. (C) The reflectance curves of the silk retreat samples. The shaded areas denote standard deviations. The curves differ most at shorter wavelengths between 300 and 470 nm where there is no overlap between the mean values reflectance values and the standard deviations. Photo credit: Alfonso Aceves-Aparicio.

**Egg-laying.** Females laid their eggs within a minimal egg-sac inside the silk retreats.

**Egg guarding.** Occasionally, spiders residing in a retreat with their eggs were seen to detect small ants approaching. They ran to the ants and physically threw them off the tree trunk before returning to their retreats.

# DISCUSSION

The aim of our study was to investigate an unusual semi-permanent retreat building behaviour in a jumping spider, a large family of spiders typically associated with a cursorial life style with only ephemeral retreats. By investigating the natural history of *Arasia mullion* we have made three major discoveries: (1) these spiders spent their entire lives on the exposed trunk's surface of a limited range of tree species; (2) they performed most aspects of their natural history in or around the silk retreats they built on the tree trunks' surfaces; and (3) during construction *A. mullion* followed a distinctive decorating behaviour using debris collected from the tree trunk that reduced the retreat's UV silk reflectance. These findings are the first description of silk decorating behaviour in jumping spiders, and highlight the unusual use of permanent silk retreats to exploit a low complexity and highly exposed environment. Our study puts a research lens on tree trunks as understudied but highly intriguing habitats.

## Living in exposed environments

Our study shows that unlike most jumping spiders, *A. mullion* is predominantly a sedentary occupant of its silk retreats on tree trunks. Whether and how these spiders move

between tree trunks is not yet known. Presumably, the relatively low structural complexity and thus, reduced availability of resources and cover on the trunks limits the prolonged presence of most animals (*Villanueva-Bonilla et al., 2020*). On the other hand, trunks could potentially provide abundant prey given the transit of insects moving between the ground and canopy. The trunks of the trees occupied by *A. mullion* offer little visual cover for the spiders. The surfaces are continuously smooth without contrasting colour patterns or crevices. Such traits pose challenging circumstances for invertebrate mesopredators such as spiders. The reduced complexity of the tree trunk surface makes the spiders vulnerable to visually oriented predators (*Gunnarsson, 1990*; *Villanueva-Bonilla et al., 2020*). Similarly, in this habitat, spiders are also more likely to be detected and avoided by their potential prey. Most animals, including jumping spiders, benefit from complex habitats and backgrounds that reduce conspicuousness (*Merilaita, Lyytinen & Mappes, 2001*). Dense foliage is used by many different spider families to physically hide their presence (*Gunnarsson, 1990*). Similarly, heterogeneous backgrounds can disrupt the visual cues used by potential prey and predators to detect jumping spiders (*Robledo-Ospina et al., 2017*).

Despite the challenges described above, other predators do reside on tree trunks. Some of these residents remain hidden behind bark or inside crevices during the day to emerge and hunt during the nights (*Cloudsley-Thompson, 1995*; *Villanueva-Bonilla et al., 2020*). Among other tree trunk residents, some astounding examples of highly cryptic appearance and posture can be found. The lichen huntsman spider (*Pandercetes gracilis*) and many species of two-tailed spiders (Hersiliidae) strikingly match the colouration patterns of the bark where they are settled. These camouflage attributes are more common in tropical forests where animals benefit from complex bark structures covered with lichen and moss (*Cloudsley-Thompson, 1995*). Unlike other tree trunk residents, *A. mullion* seems to rely on building and occupying a retreat on the surface of the trunks to allow extended trunk inhabitation.

## The use of silk retreats

The retreats built by *A. mullion* are persistent structures, unlike the usual overnight silk retreats built by most jumping spiders (*Hallas & Jackson, 1986*; *Richman & Jackson, 1992*; *Jackson & Pollard, 1996*; *Hoefler & Jakob, 2006*). Also, unlike most jumping spiders which are very active hunters, *A. mullion* perform most aspects of their life as sedentary occupants of silk retreats. These spiders were present all year round and mostly found in their silk retreats, rarely seen wandering the surface of the tree trunks (except for adult males presumably searching for females during mating season). This unusual strategy was adopted by hatchlings and continued until maturity where females laid their egg sacs inside the retreats. As spiders grew over time, they not only built new, larger retreats but engaged in dynamic patterns of retreat occupancy, repair and defence. As retreats can potentially outlive any spider, abandoned silk retreats were re-occupied by spiders at any given time. Although these observations are limited in that we cannot estimate the number of spiders moving between silk retreats due to the lack of individual spider tracking, we inferred high dynamism of retreat occupancy from the observed natural history events.

*Aceves-Aparicio et al. (2018)* recorded similar patterns of silk structure reuse by spiders other than the builder during the dispersal stage of a subsocial spider, suggesting that the abandoned three-dimensional webs were used as stepping stones by males and females while new space was colonised. Additionally, the retreats of *A. mullion* were not only re-occupied at different times but were equally likely to increase or decrease in size. This indicates that spiders were actively repairing or expanding pre-existing silk structures on the tree trunks. This is similar to beavers, which constantly repair their dams to maintain the suitability of the safe space for protection and inhabitation (*Andersen & Shafroth, 2010*). At a much smaller scale, *A. mullion* dynamically occupied and repaired the available retreats in their landscape. It is likely that the construction of, and sedentary life in, the retreats counteract the lack of cover on these smooth tree trunks. Thus, each silk retreat would hold high value for the spiders as these seem more suitable spaces for them than any open space on the trunk's surface.

## A novel method of retreat decoration

The construction behaviour of *A. mullion* is a key element to its unusual natural history. Although many jumping spider species build retreats similar in shape to those made by *A. mullion*, these are usually short-lived structures, hidden from sight on the underside of leaves, beneath rocks or behind bark (*Hoefler & Jakob, 2006*; *Hill et al., 2019*). These silk retreats serve as temporary resting sites or protective structures for egg sacs (*Foelix, 2010*). Due to the exposed nature of tree trunks, building a retreat that is hidden from view is not possible. We argue that decorating the retreat with the debris is a strategy to reduce the visibility of the bright silk. The debris is collected from close to the retreat position and serves to match the appearance of the retreat to that of its background, at least to a human observer. This scraping behaviour is reminiscent to other animals that utilise environmental material for camouflage. For example, grounds nesting birds also camouflage their eggs with soil they collect from immediate surroundings of the nest (*Mayani-Parás et al., 2015*).

Decorating behaviour among non-human animals has been primarily studied in aquatic species (*Ruxton & Stevens, 2015*). For example, decorator crabs collect elements from the environment to cover themselves, likely gaining physical protection and reducing detection by predators (*Hultgren & Stachowicz, 2009*; *Ruxton & Stevens, 2015*). Similar patterns have been documented for other aquatic fauna such as sea urchins, brachyuran, hermit crabs, and caddisfly larvae (*Ross, 1971*; *Wicksten, 1986*; *Otto, 2000*; *Dumont et al., 2007*). Comparable uses of external materials for decoration have been explored for terrestrial animals, mostly among larvae of several insect species (*Nakahira & Arakawa, 2006*; *Jackson & Pollard, 2007*; *Khan, 2020*). Spider species across several families have evolved setal microstructures that render them cryptic (*Duncan, Autumn & Binford, 2007*; *Gawryszewski, 2014*) by retaining debris in the case of *Stephanopis* (Thomisidae) and sand particles in *Sicarius* (Sicariidae) and *Homalonychus* (Homalonychidae) genera. In all these instances, the decorations are added to the animal's body. The use of decorations has also been recorded on the silk snares of some spider species (*Herberstein et al., 2001*). *Cyclosa* spiders decorate their webs with debris such as prey remains. By sitting within the

decorations, the spiders successfully deflected attacks from avian predators in experimental laboratory trials (*Ma et al., 2020*). However, decorations among jumping spiders were previously unknown and remain generally unexplored among structures built by other animals (*Hansell, 2005*; *Stevens & Ruxton, 2019*).

The two main functions of constructions among animals are as protective retreats or as traps used by predatory species. Retreats protect against physical (temperature, humidity, rainfall) or biological hazards (predation or parasitism), while traps facilitate foraging by detecting, slowing or restraining potential prey (*Hansell, 2005*). Whether either of these constructions is actively decorated to reduce detection or recognition by unwanted observers has different implications. By definition, traps should not be detected or recognised by the intended target, thus these are commonly highly inconspicuous. On the other hand, the decoration of homes (retreats, nests or burrows) for protection has often been suggested but commonly lacks evidence (*Ruxton & Stevens, 2015*). Splitting these two functions is often problematic among spiders as their silk snares might act simultaneously as a retreat and as a trap. The retreat of *A. mullion* itself is not a trap, but it could contribute to the spider's hunting strategy by reducing its exposure to potential prey. Our study shows that the hunting strategy and virtually every behavioural trait is closely related to its decorated silk home, thus adding to the scarce evidence of the active hiding of animal-built structures.

As decorations were always present in the silk retreats, it is likely that these provide significant advantages over non-decorated silk in this habitat. The behaviour of *A. mullion* showed close dependence on its decorated retreat for both predator and defence strategies. The spiders waited for prey to approach them while remaining partially covered by the silk retreat. When the potential prey escaped these attacks, the spiders quickly returned to cover within the retreats and re-settled for further capture attempts. At the same time, when approached by potential threats, the spiders moved inside the silk retreats. The comparison between non-decorated and decorated silk retreats gives us insights into how the debris cover might affect visibility. Spider silk visibility is a common constraint on silk structures in both foraging and anti-predator contexts. A web should be either attractive or not visible to the targeted prey, while to avoid predation it should be invisible or even act as a deterrent (*Zschokke, 2002*). How silk is perceived by predators and prey depends on their respective visual systems. Insects, lizards and birds are ecologically relevant observers of spider webs and can perceive light in the ultraviolet (UV) spectrum (*Blackledge & Wenzel, 2000*). Thus, if the debris reduced the overall reflectance of UV, it might also reduce the overall visibility of the silk retreats on the tree trunks. However, this remains to be explored as our study has not attempted to assess how *A. mullion* and their retreats are visually perceived. Further studies should especially consider visual systems with the capacity to detect UV light reflectance.

Alternative functions of external materials used in silk webs have been explored in multiple systems (*Herberstein et al., 2001*). A strengthening function has been suggested for the silk decorations added to orb webs to make these more stable (*Robinson & Robinson, 1970*). However, evidence is inconclusive and requires further study

(*Herberstein et al., 2001*). For *A. mullion*, strengthening might not be required for stabilisation but for direct deflection of attacks upon contact with intruders (other spiders) or predators. However, our observations have not identified potential predators of these spiders or their interaction with the silk retreats. The thermoregulation function of disc-shaped decorations in orb webs was tested in high-temperature environments and under direct exposure to sunlight. Juvenile *Neogea* spiders reduced their body temperature by moving behind the shadow produced by the silk disc (*Humphreys, 1992*). The silk retreats of *A. mullion* might be protected from direct sunlight for most of the day below the canopy. Exploratory measurements of the temperature inside and outside empty silk retreats under direct sunlight exposure did not reveal significant differences (unpublished data, Aceves-Aparicio). However, further exploration of temperature control might be pertinent considering the spiders and their manipulation of the silk retreat openings.

Researchers regularly discover novel and exceptional natural histories among jumping spiders, such as vegetarianism, blood-feeding, and sophisticated trial-and-error signalling behaviours (*Jackson & Wilcox, 1998*; *Jackson, Nelson & Sune, 2005*; *Meehan et al., 2009*). Our study of the natural history of *A. mullion* reveals a novel behavioural strategy on tree trunks. The construction of permanent retreats that are decorated with debris likely enables hunting of prey and hiding from predators in a highly exposed environment. Hence, these spiders engineer their own exclusive niche in a challenging habitat. The strategy of *A. mullion* is unusual amongst jumping spiders, or indeed many animals that build retreats. Thus, this system provides a novel avenue to approach and bridge studies regarding the functions of animal constructions and concealment strategies.

## CONCLUSIONS

In this study we have made fascinating observations that to our knowledge are unique among jumping spiders. Most other species of jumping spiders are itinerants on the surface of tree trunks, however the prolonged occupancy of *A. mullion* seems to provide benefits. The exploitation of tree trunks as a foraging arena is advantageous as tree trunks can concentrate insect traffic. We hypothesise that active search for prey on tree trunks is enhanced by hunting from a silk retreat, which would also protect the occupant from predators and unfavourable environmental conditions. *Arasia mullion* spends its entire life-cycle in these semi-permanent silk retreats. Thus, *A mullion* spiders behave like sit and wait predators, unlike most jumping spiders. We also discovered a novel decoration behaviour that may facilitate the exploitation of a highly exposed and challenging habitat–the tree trunk. The debris decorations applied to the silk retreats not only appear to match the tree trunk (at least to human observers) but also greatly reduce the UV reflectance of the silk.

Further studies are required to explore the effect of tree trunk variation on the appearance of the silk retreats and how this might affect detection by visually oriented prey and predators. Similarly, possible sources for trade-offs of this strategy are yet to be identified.

### Diversity and inclusion statement

We unreservedly support equity, diversity and inclusion in science (*Rößler, Lotters & Da Fonte, 2020*). The authors come from different countries (Mexico, Germany, Austria and Australia) and represent different career stages (undergraduate student, PhD student, early career researcher to professor). One or more of the authors self-identifies as a member of the LGBTQI+ community. While citing references scientifically relevant for this work, we actively worked to promote gender balance in our reference list. We ensured sex balance in the selection of non-human subjects as we sampled both male and the female spiders.

## ACKNOWLEDGEMENTS

We acknowledge the *Wallumattagal clan of the Dharug nation* as the traditional custodians of the Macquarie University land. We are grateful to Liliana Cadavid Florez for her assistance during spectral measurements and to Lizzy Lowe and Thomas White for their constructive and thoughtful comments. We thank two anonymous reviewers for their helpful and constructive comments.

### Funding

Alfonso Aceves-Aparicio was supported by a Cotutelle international Macquarie University Research Excellence Scholarship (CTiMQRES). The funders had no role in study design, data collection and analysis, decision to publish, or preparation of the manuscript.

### Grant Disclosures

The following grant information was disclosed by the authors:
Cotutelle international Macquarie University Research Excellence Scholarship.

### Competing Interests

The authors declare that they have no competing interests.

### Author Contributions

- Alfonso Aceves-Aparicio conceived and designed the experiments, performed the experiments, analyzed the data, prepared figures and/or tables, authored or reviewed drafts of the paper, and approved the final draft.
- Donald James McLean conceived and designed the experiments, performed the experiments, analyzed the data, prepared figures and/or tables, authored or reviewed drafts of the paper, and approved the final draft.
- Zoe Wild performed the experiments, analyzed the data, authored or reviewed drafts of the paper, and approved the final draft.
- Jutta M. Schneider conceived and designed the experiments, authored or reviewed drafts of the paper, and approved the final draft.

- Marie E. Herberstein conceived and designed the experiments, authored or reviewed drafts of the paper, and approved the final draft.

## Data Availability

The code and necessary data files to reproduce the results and figures are available on GitHub and Zenodo:

-https://github.com/PonchoAceves/Blankie-natural-history

-PonchoAceves. (2021). PonchoAceves/Blankie-natural-history: Submission (1.0). Zenodo. https://doi.org/10.5281/zenodo.4765276.

## Supplemental Information

Supplemental information for this article can be found online at http://dx.doi.org/10.7717/peerj.12839#supplemental-information.

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
