# Peer review of "Novel decorating behaviour of silk retreats in a challenging habitat"

_PeerJ, doi:10.7717/peerj.12839_

## Round 0.1 · original submission · Major Revisions

It is not clearly stated whether you observed each spider/web once whether there were repeated measurements. Please state this in the Methods. If the latter, you will need to reconsider your chosen statistical models. I also recommend changing the figure legend in Figure 4 to a vertical orientation as horizontal presentation is not clear.

Reviewer 1 ·

Basic reporting

I really enjoyed reading this manuscript. It was clearly written, well organized, and well referenced. The only place I thought might need more references was lines 70-73 to support those statements.

Experimental design

This is an observational study to learn more about the retreat building and foraging behavior of a newly discovered species of jumping spider. The extensive observations and measurements were appropriate to achieve this goal.

Validity of the findings

I loved learning more about these spiders and how they survive on smooth tree bark. Most of the conclusions reached in the Discussion section are consistent with the data presented. The only conclusion that I was less convinced of was "spiders were not transient inhabitants" (line 303). While I agree that the data is supportive of the conclusion that the spiders live and forage on exposed trunk surfaces, it still seems possible that they could move among trees. Even jumping spiders could balloon, or could crawl on the ground to a nearby tree.

Additional comments

Fun read!

Reviewer 2 ·

Basic reporting

The manuscript “Novel decorating behaviour of silk retreats in a challenging habitat” describes the retreat building of the jumping spider A. mullion and provides details of their life history based on observational data. In general, the manuscript can benefit from some re-writing (see bellow and line comments).

The abstract could benefit from a reduction in the amount of background information and the addition of more detail regarding the work and main findings. I suggest you condense the first 3-4 sentences (lines 10-19), state your hypothesis, and add the main findings.

Introduction lacks focus and the research question is not well defined. The concept of habitat structural complexity should be mentioned in the second paragraph, as it stands there is an abrupt change of topics with no linkage. I suggest rewording the examples of orb-web spiders that build retreats in terms of habitat complexity. Are these webs in a highly complex environment and thus building a retreat increases survival or allows for expansion into more complex habitats. More detail on the species used in the study will be a good addition in the introduction and the section of the methods can be shortened. Why was this species chosen? Distribution? Other than it resides on a particular tree? Is it the only species of jumping spiders on the tree? Such description could be added in line 79.

The Discussion can benefit from more cited work as well as more emphasis on why the current work is significant to the reader. There should also be a discussion regarding the role of trunk scrapping in retreat decoration.

The conclusion section reads like the last paragraph of the discussion instead of a conclusion. The conclusion should summarize the work, major findings and how they support your hypothesis, etc.

Experimental design

While the findings are novel and interesting to the scientific community, the research question is not well defined. As mentioned elsewhere, the introduction and the abstract need to be reworded to clearly state the hypothesis and how the data is/or not supporting it.

Validity of the findings

See other comments.

Additional comments

After the first use of the species name, abbreviate the genus name. Check the manuscript for consistency. To aid with clarity, please use commas to separate three or more items. Figure 1: Species names should be in italics

Video S1: If possible, I would suggest that the video is presented in full instead of cropped to see the complete sequence in retreat construction. You can change the video speed to match the desired length.

Line Comments:
Line 53: If the common name for spiders in the genus Seothyra is known please include it (e.g. sand-dwelling/spoor).

Line 61: comma after flexibility

Line 61-64: this sentence is confusing as written. Please revise and rewrite. Maybe something like: “Considering the functionality, flexibility, and broad benefits of web building, it is surprising that some groups have abandoned capture webs and instead hunt by vision and ambush (e.g. jumping spiders).” I suggest the edit because technically speaking active hunting spiders have not lost the ability to build webs, they still have the machinery (silk-producing glands) to do so.

Line 65-66: replace parenthesis with commas

Line 68: replace “Because” with “Given the astounding acute vision of jumping spiders,”

Line 71: what other types of active outbound journeys are known for spiders? If none, consider explicitly stating searching for prey and mates.

Line 71: Can you provide a reference for this statement as well as more detail? Which jumping spiders? Or generalize the statement to say that there are jumping spiders that do build silk retreats.

Line 79: add (Zabka, 2002) after the species name

Line 125: change “was” for “were”

Line 132: add “R” after conducted. Alternatively, the sentence can be written as: “All statistical analyses, including log transformations to meet statistical assumptions, were conducted in R (R Core Team, 2021).”

Line 136: The species of spiders should be included. It is unclear what type of spiders were used for the comparison.

Line 149: why were the measurements restricted to 300 to 700 nm?

Line 157: comma after “decorate”

Line 181: add “(Video S1)” to the end of the paragraph.

Line 190: Is there a time frame for retreat construction? An average time should be added here.

Line 201: Remain the reader of the importance of the observations by including that this is a species of jumping spiders. Something like: “The jumping spider A. mullion spend their entire life on the surface of tree trunks. Unlike other salticids, A. mullion spiders stayed in their silken retreat and were rarely … trunks.”

Line 210: use instead of “run”

Lines 233-236: these sentences are confusing to me, and I think they would benefit from a rewrite. You recorded 94 retreats in 2018 with 1 new retreat added in the second survey? Add the date for clarity. Being more explicit about the dates could help clear this confusion. For example: The variation in retreat persistence was 47-82%, but for which dates? All surveys? Half? You could combine the decline of retreat persistence (8.75% from XX) to coinciding with tree shedding (Lines 235-238).

Lines 254- 260: Spectral reflectance section. As per my previous comment, the reason why the wavelengths of 300 and 700 nm were chosen needs to be included. Is this change in mean reflectance significant?

Line 268: Would it be appropriate to label each retreat entrance to note differences in usage?

Line 293: The discussion would benefit from a few introductory sentences reminding the reader what the paper is about and why the work is significant. I would suggest adding details such as the fact that A. mullion is a jumping spider, part of one of the largest families of spiders. Jumping spiders usually have a cursorial lifestyle and are active predators instead of a sit and wait, salticids are known for jumping at prey/or as defense, hence the name. They are known for their large eyes and great vision which makes them very good active diurnal hunters. Some of these details should also be included in the introduction.

Line 297: add “and” before 3)

Lines 203- 317: Are there trade-offs that could account for A. mullion strategy such as size, prey abundance, predation, clutch size, survival rate, etc.? Are they the only species of spiders to use this tactic? Only salticid? Why is living on the tree trunks and relying on silk retreats significant to A. mullion?

Line 378: Should Stephanopsis by Stephanopis? Also, make sure to use italics font.

Line 421: this statement needs a citation and maybe an example.

---

## Round 0.2 · accepted · Accept

I would suggest acknowledging the two reviewers that spent considerable time and effort improving your manuscript. Also, please pay attention to your use (or lack thereof) of the Oxford comma; its absence leads to some confusing sentences in your manuscript.